# Stuffed Mamba:
# Oversized States Lead to the Inability to Forget

**Yingfa Chen, Xinrong Zhang, Shengding Hu, Xu Han**[*]**, Zhiyuan Liu**[*]**& Maosong Sun**
Department of Computer Science and Technology,
Tsinghua University,
Beijing, China
yingfa-c24@mails.tsinghua.edu.cn, {han-xu,liuzy}@tsinghua.edu.cn

## Abstract

Recent advancements in recurrent architectures, such as Mamba and RWKV, have showcased strong language capabilities. Unlike transformer-based models, these architectures encode all contextual information into a fixed-size state, leading to great inference efficiency. However, this approach can cause information interference, where different token data conflicts, resulting in performance degradation and incoherent outputs beyond a certain context length. To prevent this, most RNNs incorporate mechanisms designed to "forget" earlier tokens. In this paper, we reveal that Mamba-based models struggle to effectively forget earlier tokens even with built-in forgetting mechanisms. We demonstrate that this issue stems from training on contexts that are too short for the state size, enabling the model to perform well without needing to learn how to forget. Then, we show that the minimum training length required for the model to learn forgetting scales linearly with the state size, and the maximum context length for accurate retrieval of a 5-digit passkey scales exponentially with the state size, indicating that the model retains some information beyond the point where forgetting begins. These findings highlight a critical limitation in current RNN architectures and provide valuable insights for improving long-context modeling. Our work suggests that future RNN designs must account for the interplay between state size, training length, and forgetting mechanisms to achieve robust performance in long-context tasks.

## 1 Introduction

Transformer-based large language models (LLMs) (Achiam et al., 2023; Dubey et al., 2024) have shown impressive capabilities in processing very long sequences (Gemini Team et al., 2024; MiniMax et al., 2025). However, these models incorporate self-attention (Vaswani et al., 2017) whose complexity scales quadratically with sequence length, making long-context processing costly. In contrast, recurrent neural networks (RNNs) (Bengio et al., 1994) have a fixed-size contextual memory. Thus, their per-token time and space complexities are constant and they are much more efficient for long sequences. Despite this advantage, their effectiveness in modeling long contexts remains underexplored. Most recent state-of-the-art (SOTA) RNNs, such as Mamba-1 and Mamba-2 (Gu & Dao, 2023; Dao & Gu, 2024), GLA (Yang et al., 2024a), and RWKV (Peng et al., 2024a) are trained on context lengths below 10K, and existing works have shown that their performance degrades sharply when the context length exceeds the model's training length[1] (Ben-Kish et al., 2024; Zhang et al., 2024a; Waleffe et al., 2024).

In this paper, we analyze the factors that cause the inability of Mamba-based models to handle contexts longer than the training length. By inspecting memory retention strength and modifying the forgetting mechanism, we discover that the performance drop is caused

---

[*]Corresponding Authors.
[1]Throughout this paper, "training length" refers to the context length used during training.

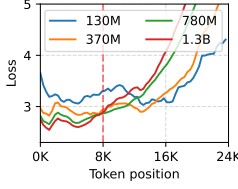
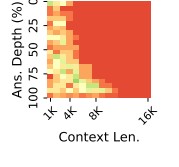 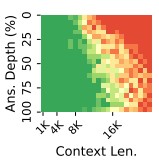 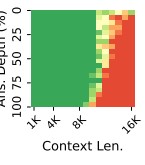 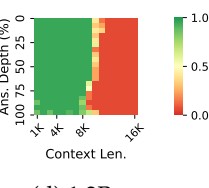

(a) 130M  (b) 370M  (c) 780M  (d) 1.3B

Figure 1: The LM loss of Mamba-2 as a function of token position. The training length is 8K.

Figure 2: The accuracy of Mamba-2 on the passkey retrieval task. "Ans. Depth" refers to the passkey position divided by the context length.

by the *inability to forget earlier tokens*, although Mamba has a built-in forgetting mechanism. Insufficient forgetting leads to interference between multiple token representations, causing faulty memory recall and, ultimately, performance degradation in longer contexts. We provide two lines of evidence for this discovery: (1) The first token's retention strength is very high throughout its training context window. (2) Artificially inducing forgetting via interventions of the state's update rule can mitigate this performance degradation.

We hypothesize that the inability to learn an effective forgetting mechanism is due to **state overparameterization**—where the model's state is excessively large, allowing it to minimize language modeling loss without much forgetting. Two key pieces of evidence support this hypothesis. (1) Initially, the model demonstrates robust forgetting, retaining only the last $k$ tokens and forgetting earlier ones. However, as training progresses, the model's ability to forget diminishes while its recall of contextual information improves, resembling overfitting. This suggests that the model increasingly attempts to retain all available information within the context. (2) We observe that forgetting occurs only when the training context length exceeds the state's capacity to retain all information, forcing the model to forget less relevant details. Notably, larger states require longer training context lengths to effectively learn and implement forgetting.

We next investigate the **minimum training length** required for Mamba-2 to effectively learn forgetting and the **maximum context length** in which the model can recall information. First, by varying model sizes and training lengths, we observe that the training length threshold scales linearly with the state size, confirming that forgetting only occurs when the training length exceeds the model's state capacity. Second, while this threshold represents the point at which the contextual information exceeds the state's capacity, we demonstrate that the model can still recall tokens beyond this context window. Evaluation on passkey retrieval (Mohtashami & Jaggi, 2023)—a simple retrieval task—shows that the maximum context length with perfect retrieval accuracy scales exponentially with state size. Notably, with continued pre-training, Mamba-2 with 370M parameters achieves near-perfect retrieval on a 256K context length, outperforming similarly sized transformer models. These findings suggest that current training lengths for RNN models may be suboptimal and underscore the potential of RNN-based architectures for modeling long-context sequences.

This paper is structured as follows. Section 2 describes the Mamba-2 architecture and provides evaluation results showing its inability to generalize beyond its training length. Section 3 provides arguments for the importance of forgetting and provides evidence showing that the model has failed to learn a robust forgetting mechanism. Section 4 presents a high-level explanation for why Mamba-2 has failed to learn how to forget. Finally, in Section 5, experiments are conducted to verify the claims and provide important conclusions for training long-context recurrent models.

The main findings of this paper can be summarized as follows:

**The inability to forget.** We discover that Mamba-2, and RWKV-6 (some SOTA RNNs) do not know how to robustly forget earlier information to avoid memory overload. This causes performance degradation for contexts longer than the training length.

**State overparameterization.** We provide overparameterization as a plausible explanation for the inability to forget and provide empirical evidence for this hypothesis.

**Minimum training length of forgetting.** In alignment with the state overparameterization hypothesis, we empirically discover that for any state size, there exists a training length threshold where the Mamba-2 learns forgetting if and only if the training length is above that threshold. We also find that this relationship is linear.

## 2 Preliminaries

In this section, we first describe the Mamba-2 architecture and corresponding notations. Then, we evaluate Mamba-2 on language modeling and passkey retrieval with context lengths exceeding their training length to illustrate the consequence of the inability to forget.

Most experiments in this study focus on Mamba-2 (Dao & Gu, 2024) because it has shown strong capabilities on several tasks and has publicly available checkpoints of multiple sizes, allowing us to explore the relationship between state sizes and length limits. Moreover, it is more widely studied than other RNNs, making it easier to use existing works as a reference.

### 2.1 Mamba-2

The Mamba-2 architecture consists of $L$ layers, each consisting of $H$ heads computed in parallel. The layer's output is the sum of the heads' outputs. Let $u_t \in \mathbb{R}^d, y_t \in \mathbb{R}^P$ denote the input and output vectors of the layer at $t$ time step. The computation at $t$ time step for each head can be formulated as follows:

$$y_t = C_t h_t \in \mathbb{R}^{1 \times P} \qquad \text{(Query rule)} \qquad (1)$$

$$h_t = h_{t-1} \underbrace{\alpha_t}_{\text{Decay}} \underbrace{+ \overline{B}_t x_t}_{\text{Insertion}} \in \mathbb{R}^{N \times P} \qquad \text{(Update rule)} \qquad (2)$$

where $C_t \in \mathbb{R}^{P \times N}, \overline{B}_t \in \mathbb{R}^{N \times 1}, x_t \in \mathbb{R}^{1 \times P}, \alpha_t \in \mathbb{R}$ are functions of $u_t$, $d, N, P$ are hyperparameters, denoting the hidden dimensionality, state dimension, and head dimension, respectively, and $h_t$ is the $t$-th *recurrent state*. Eq 1 and 2 are called the "query rule" and "update rule" because they determine how memory is queried from the recurrent state, and how the state is updated.

The other variables are parameterized as follows:

$$\overline{B}_t = B_t \Delta_t \in \mathbb{R}^{N \times 1} \qquad (3)$$

$$\alpha_t = \exp(-\Delta_t \exp(A)) \in \mathbb{R} \qquad (4)$$

$$\Delta_t = \text{Softplus}(u_t W_\Delta + b_\Delta) \in \mathbb{R} \qquad (5)$$

where $B_t \in \mathbb{R}^{N \times 1}$ is a function of $u_t$ and $A, b_\Delta \in \mathbb{R}, W_\Delta \in \mathbb{R}^{d \times 1}$ are trainable model parameters. Appendix A presents more details on the model. Notably, Mamba-2's update rule is similar to many existing RNNs (Peng et al., 2024a; Sun et al., 2023; Yang et al., 2024a). Thus, some conclusions/insights may apply to other architectures. We leave such exhaustive ablation studies for future work.

Importantly, $h_t$ is the contextual memory representation that stores information from all tokens up to $t$. $\alpha_t \in (0, 1)$ is the *memory decay* multiplier that controls the strength of forgetting. Past information is completely forgotten when $\alpha_t \to 0$ and completely retained with $\alpha_t \to 1$. In this paper, we refer to $\alpha_t$ as the **memory retention strength**.

### 2.2 Length Generalization Failure of Mamba-2

**Language Modeling** Figure 1 shows the language modeling loss of Mamba-2 as a function of token position. The result shows that Mamba-2 models suffer great performance degradation when the context length is much longer than their training lengths. Furthermore, we find that larger models have worse length generalization abilities.

**Passkey Retrieval Evaluation**   Language modeling may not reflect downstream capabilities, thus, we also evaluate Mamba-2 recall ability on the passkey retrieval task (Mohtashami & Jaggi, 2023; Zhang et al., 2024a). It is a widely-used simple synthetic task where a model is prompted to recall a 5-digit *passkey* from a lengthy context.

The passkey retrieval result is reported in Figure 2. We find that Mamba-2 (except for the smaller 130M checkpoint) has near-perfect retrieval accuracy within 8K tokens, but poor or even zero accuracy on sequences longer than 16K, regardless of model sizes. This behavior is unexpected because the update rule (Eq. 2) has a stable exponential memory decay (it converges to a constant value if the variables are fixed). Therefore, we expect RNNs of such form to have a good retrieval accuracy on the last $k$ tokens, and tokens earlier than that are forgotten. However, when the context is too long, Mamba-2 fails to even recall very recent tokens. This implies that **the limitation is not the inability to retain memory about the answer, but the inability to forget irrelevant past tokens**.

More experimental details and the results of some other recurrent architectures can be found in Appendix B and D.

## 3   The Inability to Forget

In this section, we argue that Mamba-2's length generalization failure can be attributed to its inability to forget contextual information. We first provide arguments for the importance of forgetting. Then, we present empirical evidence to verify that the model has not learned a robust forgetting mechanism. Furthermore, we show how the inability to forget is manifested in the statistics of the recurrent state.

### 3.1   The Importance of Forgetting

The fact that the model fails to retrieve tokens at any position when the context length exceeds a certain threshold has one critical implication: the existence of earlier tokens damages the model's ability to recall both earlier and more recent tokens. This is because the model's state $h_t$ at time step $t$ can be formulated as a weighted sum of past information:

$$h_t = \sum_{i=1}^{t} \alpha_{i:t} \overline{B}_i x_i, \quad \alpha_{i:t} = \left( \prod_{j=i}^{t} \alpha_j \right) \in (0, 1) \tag{6}$$

When we try to retrieve the memory inserted at time step $s$, we would query the state with $C_t = \overline{B}_s$, which returns:

$$y_t = C_t \sum_{i=1}^{t} \alpha_{i:t} \overline{B}_i x_i = \alpha_{s:t} (C_t \overline{B}_s) x_s + \underbrace{\sum_{i \neq t} \alpha_{i:t} C_t \overline{B}_i x_i}_{\text{Retrieval error}}, \tag{7}$$

When all $\overline{B}_i$ are mutually orthogonal, querying the memory with $C_t = \overline{B}_s$ returns a scaled version of $x_s$. However, as the context length increases, $B_s$ cannot be mutually orthogonal, in which case multiple memory entries interfere and cause retrieval errors. A small error may not affect the final output because subsequent calculations may tolerate these errors. However, if there are many preceding tokens, then this retrieval error can be too large. To optimize retrieval accuracy, the model needs to produce a small enough decay $\alpha_t$ to diminish the inference from earlier tokens, which can be viewed as forgetting them.

### 3.2   Evidence for the Inability to Forget

Here, we provide empirical evidence that confirms that Mamba-2 has failed to learn how to forget past information.

#### 3.2.1   *Evidence 1: High Retention of the First Token*

Based on Eq. 6, we can view $\alpha_{i:t}$ as the memory strength of the $i$-th token at $t$ time step. The retention strengths of earlier tokens are always smaller than those of more recent tokens.

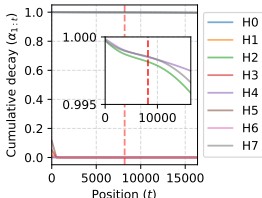
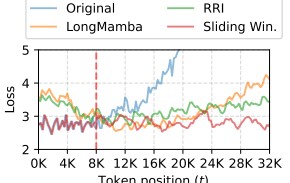
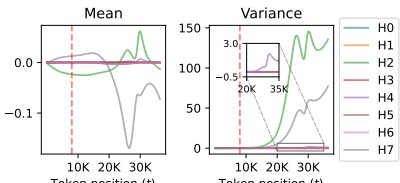

Figure 3: The retention strength of the first token ($\alpha_{1:t}$) over time. Each curve represents a head.

Figure 4: LM loss of Mamba-2 370M at different positions when inducing forgetting (see Section 3.2.2).

Figure 5: The mean and variance of the first 8 heads in layer 38 of Mamba-2 370M. It exhibits a clear explosion when $t$ is greater than the training length.

We find that some heads have a very strong inclination toward retaining all information within the training length. As an example, Figure 3 shows the cumulative decay of the first token in the first eight heads of the 38th layer, and three of the heads have a memory retention strength over 0.997[2] at $t$=8K. Similar observations can be found in other heads and in other layers as well. This implies that the model has not learned to forget information (by producing a smaller $\alpha_j$), but it still has decent language modeling capabilities because the information of 8K tokens is typically not enough to overload the memory.

### 3.2.2 *Evidence 2: Inducing Forgetting Can Improve Length Generalization*

Here, we demonstrate that artificially inducing more forgetting *without training* can improve performance by reducing past memory interference.

**Reduced Memory Retention and Insertion (RRI)**   This method assumes that $\alpha_t$ and $B_t$ control the memory retention and insertion strength, respectively. We scale them with a multiplier smaller than 1. The actual multipliers used are 0.9999 for $\alpha_t$ and 0.75 for $B_t$, which is chosen by validation using average loss on pre-training data with 32K context length.

**Sliding Window**   We can utilize the fact that the state $h_t$ can be written as a weighted sum (Eq. 6) to simulate a sliding window mechanism without re-processing from the start of the window at every step. Let $w \in \mathbb{N}$ denote the window size and $h_t^{(r)} \in \mathbb{R}^{N \times P}$ denote the hidden state when applying the model on the last $w$ tokens at time step $t$. We can then compute $h_t^{(r)}$ exactly as the difference between two states:

$$h_t^{(r)} = \sum_{i=t-r+1}^{t} \alpha_{i:t} \overline{B}_i x_i = \sum_{i=1}^{t} \alpha_{i:t} \overline{B}_i x_i - \alpha_{t-r+1:t} \sum_{i=1}^{t-r} \alpha_{i:t-r} \overline{B}_i x_i = h_t - \alpha_{t-r+1:t} h_{t-r} \tag{8}$$

During streaming generation, we only have to maintain $(h_{t-1}, h_{t-r}, \alpha_{t-r+1:t})$[3], and advance each of them in parallel. However, directly computing $\alpha_{t:t-r}$ may suffer from instability due to floating-point imprecision. Therefore, we maintain $\Delta_{t-r:t} = \sum_{i=t-r}^{t} \Delta_t$ instead, and re-compute $\alpha_{t-r:t} = \exp\left(-\Delta_{t-r:t} \exp(A)\right)$ at every step, which incurs minimal computational cost. This method can be applied to all RNNs that can be written as a weighted sum.

**Result**   From Figure 4, one can see that the original model has the worst length generalization abilities. LongMamba (a length extrapolation method) (Zhang, 2023) and our methods for inducing forgetting can alleviate this generalization failure, although LongMamba and RRI compromise short-context performance due to weaker memory insertion. This further confirms the fact that memory over-retention is the culprit of this degradation.

---

[2]This is a cumulative product, so the decay at each time step is even closer to 1.

[3]We also have to cache the last $r$ token IDs, but their size is negligible compared to $h_{t-1}$ and $h_{t-r}$

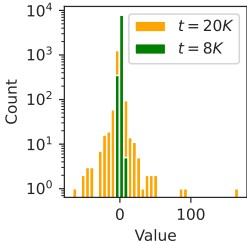

Figure 6: The distribution of the channels in an exploding state (head 2 in layer 38) at two different time steps.

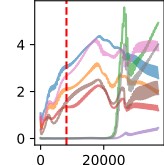 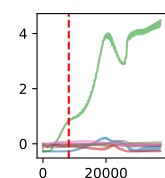 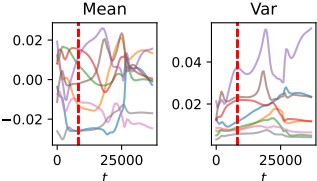

(a) $\Delta_t$ over time. Each curve represents a head.

(b) $B_t$ over time. Each curve represents a channel.

(c) $x_t$ over time. Each curve represents a head.

Figure 7: The value of various components in the update rule ($\Delta_t$, $B_t$, and $x_t$) on some heads with large retention values in the 38th layer in Mamba-2 370M. The red dotted line indicates the training length.

### 3.3 The Manifestation of Over-Retention

We also examine how the inability to forget is manifested in the state's values. Since the recurrent state's dimensionality does not change over time, the sharp change of behavior during length generalization must be a result of a change in the state's distribution. For reproducibility and better visualization, we use the "newlines" prompt (string with only "\n") and inspect the statistics of the recurrent states of every head in Mamba-2 370M[4] and find that the mean and variance of some heads change sharply when the context length exceeds the training length. One example is shown in Figure 5. Appendix G reports the statistics of every layer. The state at $t = 20K$ of one head with exploding variance is shown in Figure 6. From it, we discover that this variance explosion can be largely attributed to a few outlier channels while most channels are relatively stable.

## 4 State Overparameterization

Here, we present a high-level explanation for why Mamba-2 over-retains memories: **state overparameterization**. The state is excessively large for the training length, allowing the model to achieve strong language modeling performance without learning how to forget when the state is overloaded with memories.

To support this hypothesis, we present two pieces of evidence: (1) Mamba-2 starts with the ability to forget, but slowly loses this ability as the amount of training data increases, which coincides with behaviors of overfitting, and (2) for any state size, there is a training context length threshold $T_{\text{forget}}$ such that Mamba-2 learns to forget if and only if $T_{\text{train}} > T_{\text{forget}}$, where $T_{\text{train}}$ denotes the training context length.

### 4.1 Evidence 1: More Training $\Rightarrow$ Less Forgetting

We pre-train Mamba-2 370M from scratch with $T_{\text{train}} = 512$ using the RedPajama (Computer, 2023) corpus and evaluate the intermediate checkpoints on passkey retrieval, as reported in Figure 8. It shows that the model's retrieval accuracy for contexts longer than the training length slowly decreases as we increase the amount of training data. Meanwhile, the model's accuracy for contexts shorter than the training length increases. This indicates that the model converges toward more retention and less forgetting. Since language modeling loss is only computed for tokens within the training length, this behavior is induced by minimizing loss. This reduced forgetting leads to conflicts between token representations, impairing memory recall accuracy when an excessive number of tokens are inserted.

---

[4]Similar observation can be found with any model size.

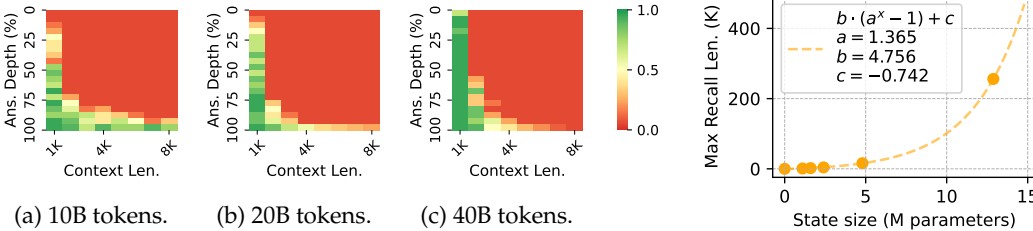

(a) 10B tokens.  (b) 20B tokens.  (c) 40B tokens.

Figure 8: Passkey retrieval results of intermediate check-points during the pre-training of Mamba-2 370M on 512 sequence length. Generalization failure only occurs in the model beyond a certain amount of training data.

Figure 9: The maximum context length of accurate passkey retrieval (i.e., $T_{\text{recall}}$ in Section 4.3) as a function of state size.

This behavior can be viewed as a kind of overfitting because the state's distribution with short contexts is not varied enough for the model to generalize to the distribution in longer contexts. In other words, the state has too many parameters for the given training length.

## 4.2 Evidence 2: Forgetting is Learned ⇔ Sufficient Training Length

We empirically confirm that larger recurrent states require longer training contexts for the model to learn to forget. This is because the model will only learn to forget when the amount of contextual information exceeds the state capacity. This hypothesis implies the following law:

> Let $N_S$ and $T_{\text{train}}$ denote the recurrent state size and training length, respectively, there exists a threshold $T_{\text{forget}}(N_S)$ such that the model learns to forget if and only if $T_{\text{train}} > T_{\text{forget}}$.

We empirically validate this by sweeping different training lengths for different state sizes, and checking whether the model has successfully learned how to forget. Concretely, we train multiple Mamba-2 with different state sizes and training lengths to find the relationship between $T_{\text{forget}}$ and $N_S$. To determine whether the model has learned robust forgetting, we feed prompts with 1M tokens to the model and check if the model's loss exceeds $2\times$ the maximum loss within $T_{\text{train}}$ tokens at any point. The loss is averaged over 128 prompts. The result is reported in Section 5.1.

## 4.3 Maximum Recall Context Length

The fact that the amount of information in $T_{\text{forget}}$ tokens exceeds the state's capacity, does not necessarily imply that the model fails to recall information beyond the last $T_{\text{forget}}$ tokens, especially when there is a clear distinction between the target information and other contextual information. Therefore, we also search for the maximum context length from which the model can accurately perform passkey retrieval. We refer to this context length as the model's *maximum recall context length*, denoted with $T_{\text{recall}}$. Similar to the previous section, we train with different lengths for different state sizes and identify the maximum context length where the model has an accuracy over 95% as $T_{\text{recall}}$. In this task, the noisy context is repetitive, thus, the amount of contextual information is largely independent of the context length. Therefore, ideally, the recall threshold should grow roughly exponentially with the state size.[5]

---

[5] If we train Mamba-2 on passkey retrieval data, the model can theoretically handle infinitely long contexts. Here, the model is only trained with the next token prediction objective, which means the model will *not* ignore the irrelevant context, and the ability to retain information for extended time emerges from language modeling.

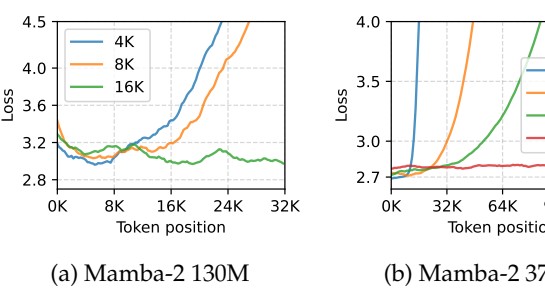

(a) Mamba-2 130M          (b) Mamba-2 370M

Figure 10: LM loss at each token position for different training lengths. Evaluated on RedPajama.

Figure 11: Minimum training length at which the model learns robust forgetting (i.e., $T_{\text{forget}}$ in Section 4.2).

## 5    Experiments

We briefly describe the data and model configurations used to identify the *forget threshold* $T_{\text{forget}}$ and *maximum recall context length* $T_{\text{recall}}$. Due to limited space, more comprehensive experimental details are reported in Appendix F.

**Data**    We start from RedPajama-V2 (Computer, 2023), an open dataset with 30T tokens from the Internet, and perform deduplication to ensure data quality and discard documents that are too short.

**Models**    We experiment with six model sizes to find the relationship between state capacity and size. For each of them, we perform an extensive search with training lengths up to 256K tokens. To save cost, we continue pre-training from three official checkpoints of Mamba-2 (130M, 370M, and 780M). They were pre-trained with 8K sequences. The other model configurations (36M, 47M, and 85M) are trained from scratch.

### 5.1    The Existence of Forget Threshold

In Figure 10, we plot the language modeling perplexity as a function of token position for Mamba-2 130M and 370M with different training lengths. We can see that for each model size, there is a training length threshold, beyond which the model has much better length extrapolation, which supports our arguments discussed in Section 4.2.

### 5.2    Forget Threshold as a Function of the State Size

Figure 11 shows the minimum training length needed for Mamba-2 to learn forgetting. The rightmost data point in the plot corresponds to Mamba-2 370M. We have confirmed that the 780M model (with a state size of 19.3M) also has poor length generalization at training lengths below 128K, but do not have enough resources to train the model beyond this length. The results establish a linear relationship $T_{\text{forget}} = 5.172 \cdot N_S - 4.469$ between the length $T_{\text{train}} = T_{\text{forget}}$ at which the model can learn robust forgetting and the state size $N_S$. The $R^2$ value is over 0.999. This indicates that **to train a Mamba-2 with robust length generalization, one should use training lengths that grow linearly with the state size.**

### 5.3    Maximum Recall Context Length as a Function of the State Size

The second plot of Figure 9 shows the recall threshold of Mamba-2 in passkey. The maximum contexts length in which Mamba-2 can accurately retrieve 5-digit passkeys is exponential concerning the state size, the function is $T_{\text{recall}} = 4.756 \cdot (1.365^{N_S} - 1) - 0.742$, with an $R^2$ value over 0.999. This is because the amount of information in the context does not increase with its length. In other words, we are storing a constant amount of information while the

number of combinations of the state grows exponentially with the number of elements. The result is very promising because, to the best of our knowledge, no previous models with less than 1B model parameters have near-perfect accuracy at this length in this task.

## 6 Related Works

**RNN-Based Language Models** This paper focuses on Mamba-2, a recurrent architecture that can be viewed as a variant of gated linear attention (Yang et al., 2024a). Many recently proposed RNNs can also be viewed as GLA variants. These include the RWKV series (Peng et al., 2023; 2024a), the Mamba series (Gu & Dao, 2023; Dao & Gu, 2024), GLA (Yang et al., 2024a), and many more (Zhang et al., 2024b; Yang et al., 2024b; De et al., 2024; Arora et al., 2024b; Orvieto et al., 2023; Sun et al., 2023). Our methods may apply to these architectures as well. Some recent/concurrent RNNs such as Gated DeltaNet (Yang et al., 2025), RWKV-7 (Peng et al., 2025), xLSTM (Beck et al., 2024), and Titans (Behrouz et al., 2024) have gone beyond a gating-based memory decay mechanism and are out of the scope of this paper.

**Length Generalization** Most SOTA language models in the last few years have been based on the transformer (Vaswani et al., 2017) architecture. These models, when using certain variants of position encoding, can process arbitrarily long sequences. However, they exhibit severe performance drops on tokens beyond the training length (Zhao et al., 2024). To alleviate this shortcoming, many works have focused on modifying positional encoding (Peng et al., 2024b; Zhu et al., 2024; Ding et al., 2024; Jin et al., 2024), some achieving training-free length generalization to certain extents (Zhang et al., 2025).

**Length Generalization of Mamba** Some prior works investigated the performance of Mamba as a function of context length (Park et al., 2024; Wen et al., 2025). Jelassi et al. (2024) empirically showed a sharp performance drop beyond the training length for Mamba on a copying task and also showed that Mamba struggles to copy from context unless its state size grows linearly with the context length. Arora et al. (2024a) discussed the associative recall abilities of transformer and some RNNs. Wang et al. (2025) is most related to our work. They discussed the issue of over-smoothing introduced by the memory decay term. In contrast, our paper explores a setting where recency may be preferred, but interference from earlier tokens damages the recall accuracy of recent tokens.

Some concurrent works have explored extending Mamba's context length by controlling the discretization term ($\Delta_t$ in Eq. 2) (Ben-Kish et al., 2024), such as dividing it by a constant to make it smaller (Zhang, 2023). This makes the memory decay factor ($\alpha_t$ in Eq. 4) closer to 1, which makes the state retain more contextual information. However, it also unnecessarily diminishes the inserted information on all tokens. Similar to the above works, this study explores the cause of Mamba-2's inability to generalize beyond its training context length and provides valuable insights into training Mamba-2 models that generalize better.

## 7 Conclusion

This paper demonstrates that while the Mamba architecture includes a memory decay mechanism, it fails to effectively learn forgetting in practice. As a result, when the context exceeds the training length, the model produces incoherent outputs. This issue arises from training with contexts that are too short relative to the state size. Empirical results show that robust forgetting is only learned when the training context length surpasses a certain threshold, which increases linearly with the state size. Notably, the model is still capable of recalling some contextual information beyond this threshold. These findings offer valuable insights into the causes and consequences of the model's inability to forget, highlighting key limitations of the Mamba architecture. Nevertheless, the insights gained from this study provide a promising foundation for improving Mamba's performance in long-context modeling, paving the way for more effective applications in tasks requiring extended context lengths.

## Acknowledgements

This work is supported by the high-quality development project of MIIT and a grant from the Guoqiang Institute, Tsinghua University.

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

# A Mamba-2 Architecture

For completeness, we give a more detailed formulation of the Mamba-2 architecture here, although we recommend the readers refer to the original paper (Dao & Gu, 2024) or a detailed blog post by the authors[6]. The model accepts a sequence of $T$ token IDs as input $\mathbf{I} = [i_1, \cdots, i_T] \in \mathbb{N}^t$, where $i_t \in \{1, 2, \cdots, V\}$, $V$ denotes the vocabulary size. It performs next token prediction by predicting the probability distribution over the vocabulary at each time step, denoted as $P \in \mathbb{R}^{T \times V}$. The model can be formulated as follows.

$$\mathbf{U}^{(0)} = \text{Embed}_{\text{in}}(\mathbf{I}) \in \mathbb{R}^{d \times T}$$

$$\mathbf{U}^{(l)} = \text{Mamba}^{(l)}\left(\text{Norm}\left[\mathbf{U}^{(l-1)}\right]\right) \in R^{d \times T}$$

$$\mathbf{P} = \text{Embed}_{\text{out}}\left(\text{Norm}\left[\mathbf{U}^{(L)}\right]\right) \in \mathbb{R}^{V \times T}$$

where $L$ denotes the number of layers, $l \in \{1, \cdots, L\}$ denotes the layer index, $\mathbf{U}^{(l)} \in \mathbb{R}^{T \times d}$ represents the input of the $l$-th layer, $\mathbf{U}^{(0)}$ represents the input of the first layer. $\text{Mamba}^{(l)}(\cdot)$ denotes the $l$-th Mamba layer, $\text{Embed}_{\text{in}}(\cdot)$ and $\text{Embed}_{\text{out}}(\cdot)$ denote the input and output embedding layers, and $\text{Norm}(\cdot)$ denotes RMS normalization (Zhang & Sennrich, 2019). $d$ denotes the number of dimensions of each token embedding. Similar to many other models, Mamba-2 ties the weight of the input and output embedding layers.

Each Mamba layer consists of $H$ "heads" that are computed in parallel. The result of which is summed together. Notably, the notations here are slightly different from Section 2 because we simplified the notations in the main content to save space. The $t$-th token ($t \in \{1, \cdots, T\}$) in a head is computed as follows:

$$y_t = W_o \text{Norm}\left(o_t^\top \odot W_{\text{gate}} u_t\right) \in \mathbb{R}^{d \times 1} \tag{9}$$

$$o_t = C_t h_t + D \odot x_t \in \mathbb{R}^{1 \times P} \tag{10}$$

$$h_t = h_{t-1} \exp(-\Delta_t \exp(A)) + \Delta_t B_t x_t \in \mathbb{R}^{N \times P} \tag{11}$$

$$C_t = \sigma(\text{Conv}(W_C u_t))^\top \in \mathbb{R}^{1 \times N} \tag{12}$$

$$B_t = \sigma(\text{Conv}(W_B u_t)) \in \mathbb{R}^{N \times 1} \tag{13}$$

$$x_t = \sigma(\text{Conv}(W_x u_t))^\top \in \mathbb{R}^{1 \times P} \tag{14}$$

$$\Delta_t = \text{Softplus}(W_\Delta u_t + b_\Delta) \in \mathbb{R} \tag{15}$$

$u_t \in \mathbb{R}^{d \times 1}$ denotes the $t$-th input representation. In other words, for the $l$-th layer, we have $\mathbf{U}^{(l)} = \left[u_1^{(l)}, \cdots, u_T^{(l)}\right]$, $\mathbf{U}^{(l+1)} = \left[y_1^{(l)}, \cdots, y_T^{(l)}\right]$, and $u_t^{(l+1)} = y_t^{(l)}$. $\text{Conv}(\cdot)$ denotes a channel-wise one-dimensional convolutional layer with a kernel size of 4, and $\sigma(\cdot)$ denotes the SiLU activation function (Elfwing et al., 2018). The result of a matrix multiplied by a scalar is the matrix with each element multiplied by that scalar.

$W_{\text{gate}}, W_x \in \mathbb{R}^{d \times P}, W_o \in \mathbb{R}^{P \times d}, W_C, W_B \in \mathbb{R}^{d \times N}, W_\Delta \in \mathbb{R}^{d \times 1}, b_\Delta, A \in \mathbb{R}$ are trainable parameters of the layer, and $P, N$ are hyperparameters. The authors call $P$ the *head dimension* and $N$ the *state size*. In practice, the weights of $W_B, W_C$ are shared among different heads.

## A.1 State Size

The authors of Mamba-2 always set $P = 64$, $N = 128$, and $H = 2d/P$. Thus, the state size of each Mamba-2 layer is $HPN = 2dN = 256d$. In transformer-based models, when using multi-headed attention, usually, the product of the number of heads $H$ and head dimension $P$ equals the hidden dimension $d$. Therefore, the KV cache of a transformer-based model is $2Td$, which means that when using the same hidden dimension, the state of a Mamba-2 layer is equal in size to a KV cache of 128 tokens. Additionally, the number of Mamba-2

---

[6]https://tridao.me/blog/2024/mamba2-part1-model/

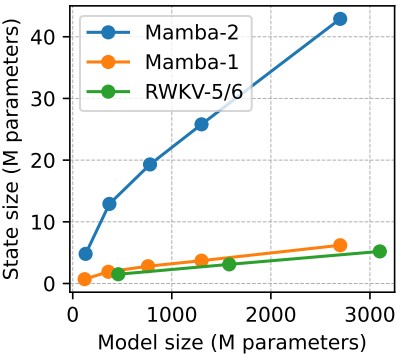

Figure 12: The relationship between state size and model size of various RNN models in this paper.

layers in a Mamba-2 model is usually twice the number of attention layers in a similar-sized Transformer model. Thus, the state size of a Mamba-2 model is equal in size to a KV cache of a vanilla Transformer model.

Compared to many other recurrent models (e.g., the RWKV series (Peng et al., 2023; 2024a), GLA (Yang et al., 2024a), and RetNet (Sun et al., 2023)), Mamba-2 does not have a state-less feed-forward network and has considerably more heads in each layer, making the state size much larger than other recurrent models. Compared to Mamba-1 (Gu & Dao, 2023), Mamba-1 uses $N = 16$, which means that the state size in Mamba-2 is 8 times larger than the state in a Mamba-1 model of roughly the model parameter count. Figure 12 shows the relationship between state size and model size of the RNN models in this study.

### A.2 Short Convolution

The $\text{Conv}(\cdot)$ function in Mamba-2 is a one-dimensional convolutional layer applied to each channel separately. For $i$-th channel, it can be formulated as follows.

$$y_{t,i} = \sum_{j=1}^{k} w_{j,i} x_{t-j,i} \in \mathbb{R}, \quad i = 1, \cdots, n_c$$

$k$ denotes the kernel size (set to 4 by default). $i$ denotes the channel index, $n_c$ denotes the number of channels. $y_{t,i} \in \mathbb{R}$ denotes the $i$-th channel of the output vector at $t$-th time step. $x_{t,i}$ represents the $i$-th channel of the input vector at $t$-th time step. $w_{j,i} \in \mathbb{R}$ denotes the $j$-th value in the convolutional kernel for channel $i$.

This model component accepts the last 4 token embeddings at the input. Therefore, it also has a state that contains information about the context, which we refer to as the *convolutional state*. To be concrete, due to information propagation through the layers, the short convolutional layer is a function of the last $4L$ tokens. For the 370M model size, this length is $4 \times 48 = 192$. Therefore, we can reasonably assume that this component contains much less contextual information relative to the recurrent state $h_t$. Thus, we have largely ignored this state in various discussions in this paper. However, we have also reported the distribution of the input to this short convolutional layer over time in Figure 17, for reference. As we can see, the convolutional state is relatively stable over time (compared to the recurrent state).

## B Passkey Retrieval Inference Parameters

Throughout the whole paper, we use greedy decoding, not just for reproducibility, but also because our preliminary results show that other decoding parameters give noticeably worse performance on passkey retrieval.

We use 32-bit floating point precision for both model parameters and activations during inference, to ensure that precision errors do not introduce noise to the result. We have conducted some preliminary evaluations with BF16 and FP16 and found that there are no noticeable differences with using FP16, but computing some activations, especially the $\Delta_t$ and $\alpha_t$ with BF16 introduces an error around 1e-3. However, the explosion of channels in the states is consistently observed despite this precision error.

### B.1 Passkey Retrieval Prompt

The prompt that we use for the passkey retrieval task is as follows, using 34847 as the passkey for example, which is adapted from existing works (Zhang et al., 2024a). We also evaluate with slight variations to the template in preliminary experiments but do not observe considerable differences in the results.

```
There is important info hidden inside a lot of irrelevant text.
Find it and memorize it.

The grass is green. The sky is blue. The sun is yellow. Here we
go. There and back again.
...
The grass is green. The sky is blue. The sun is yellow. Here we
go. There and back again.
The passkey is 34847. Remember it. 34847 is the passkey.
The grass is green. The sky is blue. The sun is yellow. Here we
go. There and back again.
...
The grass is green. The sky is blue. The sun is yellow. Here we
go. There and back again.

What is the passkey? The passkey is
```

We sweep different context lengths $T \in \{1K, 2K, ..., 256K\}$, and for each length $T$, we generate $n$ prompts with evenly distributed needle positions, i.e., the $i$-th needle ($i \in \{0, \cdots, n-1\}$) of a sample is inserted at position $T \times i/n - 1$, from the beginning.

## C Mamba-2 with Modified $\Delta_t$ on passkey retrieval

Ben-Kish et al. (2024) and GitHub user jzhang28[7] propose to improve Mamba's length generalization by reducing the value of $\Delta_t$. Ben-Kish et al. (2024) propose a heuristic method for identifying which head to modify and how to modify $\Delta_t$. However, their method requires task-dependent tweaking, so we do not consider comparing against it. jzhang28 propose to simply multiply $\Delta_t$ by a constant (they used 0.5). We apply this method and sweep different $\Delta_t$ for the best passkey retrieval performance, but got near-zero accuracy across all passkey positions and context lengths.

## D Passkey Retrieval Evaluation with Other Architectures

Here, we also evaluate RWKV-5, RWKV-6, and Mamba-1 (some popular and strong RNNs) on the passkey retrieval task. The result is reported in Figure 13, 14, and 15. We can see that length generalization failure is observed in Mamba-1, but it is less severe for RWKV-5 and RWKV-6. We hypothesize that this difference is a result of architectural differences and that the state size is smaller in RWKV-5 and RWKV-6 (see Figure 12).

---

[7]https://www.github.com/jzhang38/LongMamba

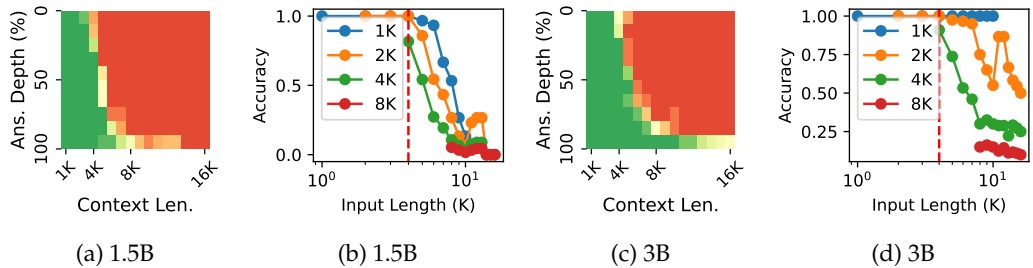

(a) 1.5B  (b) 1.5B  (c) 3B  (d) 3B

Figure 13: The performance of RWKV-5 official checkpoints on the passkey retrieval task. Each curve in (b) and (d) represents the accuracy of retrieving the needle when it is within the last $r$ tokens, with $r \in \{1K, 2K, 4K, 8K\}$.

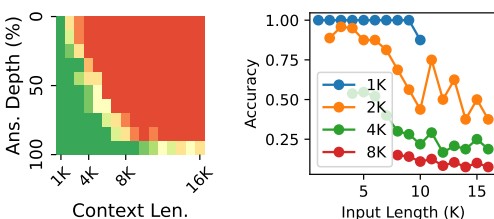

Figure 14: RWKV-6 1.6B result on the passkey retrieval task. The left plot shows the retrieval accuracy of the needle when it appears in the last $r = \{1K, 2K, 4K, 8K\}$ tokens.

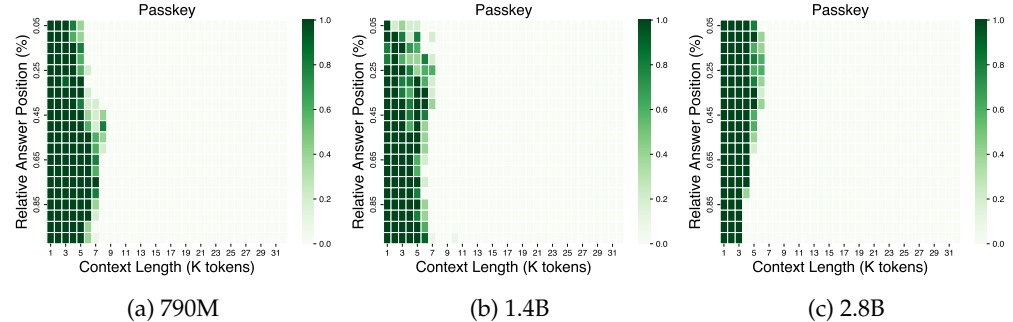

(a) 790M  (b) 1.4B  (c) 2.8B

Figure 15: The performance of Mamba-1 official checkpoints on the Passkey task. We can see a clear exhibition of the inability to forget, similar to Mamba-2.

# E  Pre-Trained Checkpoints

The pre-trained checkpoints used in our experiments are given in Table 1.

# F  More Experimental Details

**Data Processing**   To ensure that the data contains as much long-term structure as possible, we filter out sequences with less than 4K tokens. Buckman & Gelada (2024) have shown that this is critical for training effective long-context models. Although we train on sequences longer than 4K tokens, we do not use a higher length threshold because the above threshold already removes about 97.6% of the data in the original corpus. To train on longer sequences, we simply concatenate sequences and delimit them with a special EOS (End-of-Sequence) token. During evaluation, we sample documents longer than 16K tokens and concatenate them if they are not long enough.

| Model | Checkpoint URLs |
|-------|-----------------|
| RWKV-5 | `https://huggingface.co/RWKV/`
`rwkv-5-world-all-pth` |
| RWKV-6 | `https://huggingface.co/RWKV/v6-Finch-1B6-HF`
`https://huggingface.co/RWKV/v6-Finch-3B-HF` |
| Mamba-1 | `https://huggingface.co/state-spaces/mamba-130m`
`https://huggingface.co/state-spaces/mamba-370m`
`https://huggingface.co/state-spaces/mamba-790m`
`https://huggingface.co/state-spaces/mamba-1.4b`
`https://huggingface.co/state-spaces/mamba-2.8b` |
| Mamba-2 | `https://huggingface.co/state-spaces/mamba2-130m`
`https://huggingface.co/state-spaces/mamba2-370m`
`https://huggingface.co/state-spaces/mamba2-780m`
`https://huggingface.co/state-spaces/mamba2-1.3b`
`https://huggingface.co/state-spaces/mamba2-2.7b` |

Table 1: The pre-trained checkpoints used in our experiments.

**Truncated Backpropagation Through Time**    In the vanilla Mamba-2, the states are initialized to zeros for each data sample. Instead, we initialize the states as the final state of the previous sequence. This is equivalent to concatenating multiple sequences, but stopping the backpropagation of gradients at certain intervals. This technique has been shown to help extend the context length of RNNs (Yang et al., 2024a) and alleviate the memory cost of caching activations for computing gradients. We employ this technique to make the distribution of the state's initial value (e.g., the state before processing the first token $h_0$) more varied. Based on Yang et al. (2024a) and our preliminary tests, we use concatenate 12 sequences with this technique by default.

### F.1    More Hyperparameters

We use the WSD LR scheduler (Hu et al., 2024) with 10% decay steps. This scheduler is chosen because it is competitive with the commonly used cosine scheduler while allowing simple resumption from intermediate checkpoints, saving large amounts of computational resources. We report the result of the best checkpoint selection by validation on passkey retrieval.

We perform a hyperparameter search on learning rates, sweeping $\{1e-5, 2e-5, 5e-5, 1e-4, 2e-4, 5e-4, 1e-3\}$, selecting the best performing one by validation on passkey retrieval[8]. Regarding the WSD scheduler, it warms up linearly for 1000 steps and decays linearly with 50K steps. This setup is inspired by the authors of WSD (Hu et al., 2024).

Other hyperparameters are kept as similar to the original papers for Mamba-2 as possible. That means we use 0.5M tokens per batch because we found this to give more stable results for continual pre-training instead of the 1M batch size from the original paper. Training is done mainly in BF16, with some activations in FP32 (in the same manner as the official implementation). The optimizer is AdamW, with a 0.1 weight decay. Moreover, we use 1.0 gradient clipping.

All experiments are run on A800 80G, some are run with multiple nodes, and others with multiple GPUs on a single node.

### F.2    Model Configurations

For the models smaller than the 130M official checkpoint, we pre-train from scratch using the configurations reported in Table 2. We try to follow the same depth-to-width ratio found

---

[8]While the loss of many checkpoints was highly similar, their performance in passkey retrieval can differ a lot.

| Model size | State size | # Layers | Hidden dim. | # heads |
|---|---|---|---|---|
| *Official checkpoints* | | | | |
| 780M | 19.3M | 48 | 1536 | 48 |
| 370M | 12.9M | 48 | 1024 | 32 |
| 130M | 4.8M | 24 | 768 | 24 |
| *Our checkpoints trained from scratch* | | | | |
| 84.6M | 2.4M | 12 | 768 | 24 |
| 47.0M | 1.6M | 12 | 512 | 16 |
| 36.4M | 0.8M | 6 | 512 | 16 |

Table 2: The configurations of the models used in finding the passkey retrieval memory capacity as a function of the state size.

in the official checkpoints, although the ratio is not entirely consistent in those checkpoints. Hyperparameters not mentioned are kept the same as the 130M checkpoint.

## G  State Statistics over Context Length

Here, we provide a more detailed result on the inspection of state distribution over time.

Figure 16 shows the distribution of hidden state $h_t$ of the recurrent mechanism described in Eq. 2. Additionally, $B_t$, $C_t$, and $x_t$ in Mamba-2 are generated with a short channel-wise convolutional layer with a kernel size of 4:

$$B_t = \sigma(\text{Conv}[u_t W_B])$$
$$C_t = \sigma(\text{Conv}[u_t W_C])$$
$$x_t = \sigma(\text{Conv}[u_t W_x])$$

where $\sigma$ is the SiLU activation function. This function is also stateful because it operates on the last 4 tokens, therefore, we also collect the statistics of this convolutional state and report them in Figure 17. As we can see, the convolutional states are much more stable compared to the recurrent states. This is because only the last 4 tokens contribute to this state which avoids the explosion as a result of cumulative sum.

## H  Length Generalization of Other Architectures

We additionally evaluate HGRN-2 (Qin et al., 2024) and RWKV-6 (Peng et al., 2024a) on the "newlines" prompt (string with only "\n") and find that they also exhibit severe performance degradation on the "newlines" prompt. The phenomenon is less severe in RWKV-6, which concurs with our argument that with longer training length, the model will learn to more robust forgetting mechanism, thus avoiding memory overload. Perhaps surprisingly, the increase in perplexity happens considerably before the context length reaches the training length for both models. We hypothesize that this is a result of the training distribution, and that by continual training on data with more long-distance dependencies can alleviate this degradation.

## I  The "newlines" Prompt

In this paper, we collect the statistics of the state computed on a "newlines" prompt, a prompt where every token is the newline token ("\n"), as shown below.

\n\n\n\n\n\n\n\n\n\n\n\n\n\n\n\n\n...

However, we again emphasize that similar state distribution and model behavior are observed on prompts extracted from the pre-training corpus, the passkey retrieval task, or

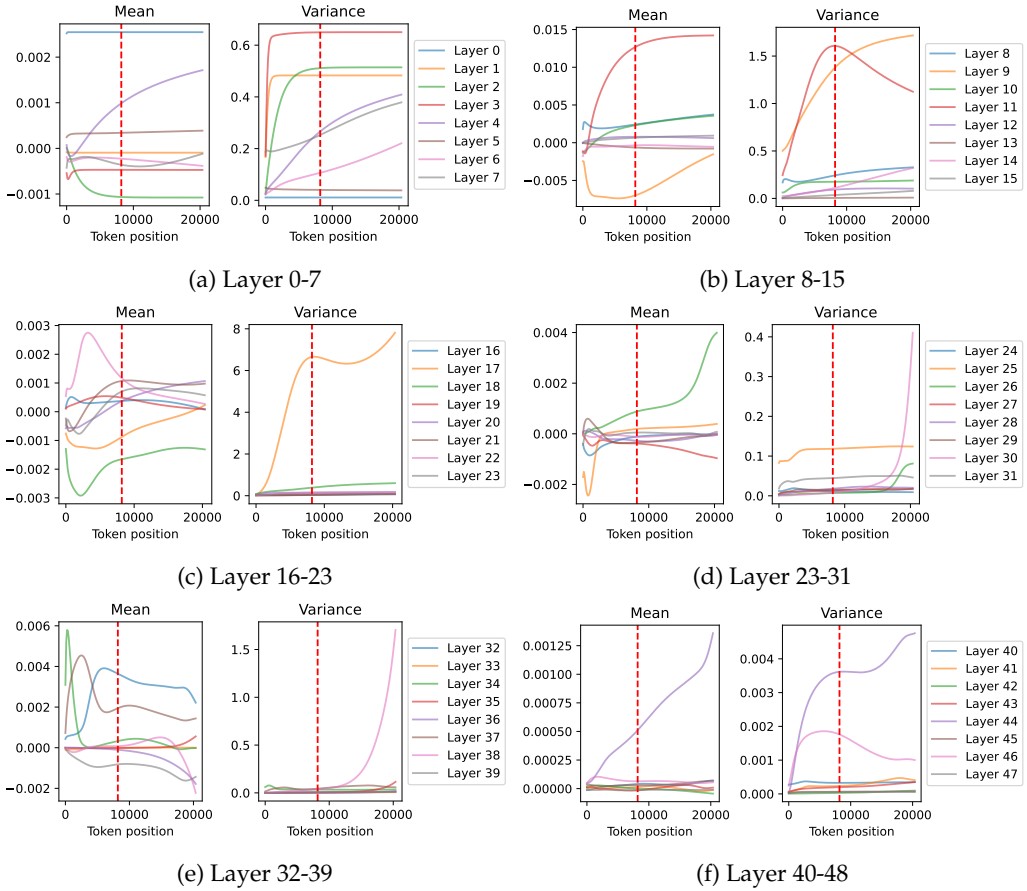

(a) Layer 0-7

(b) Layer 8-15

(c) Layer 16-23

(d) Layer 23-31

(e) Layer 32-39

(f) Layer 40-48

Figure 16: The mean and variance of the hidden state of each layer of Mamba-2 370M, computed on the "newlines" prompt (string with only "\n").

other randomly generated sequences. We have chosen the "newlines" prompt because the samples from the pre-training corpus are too short, and this prompt produces the most consistent and smooth layer statistics.

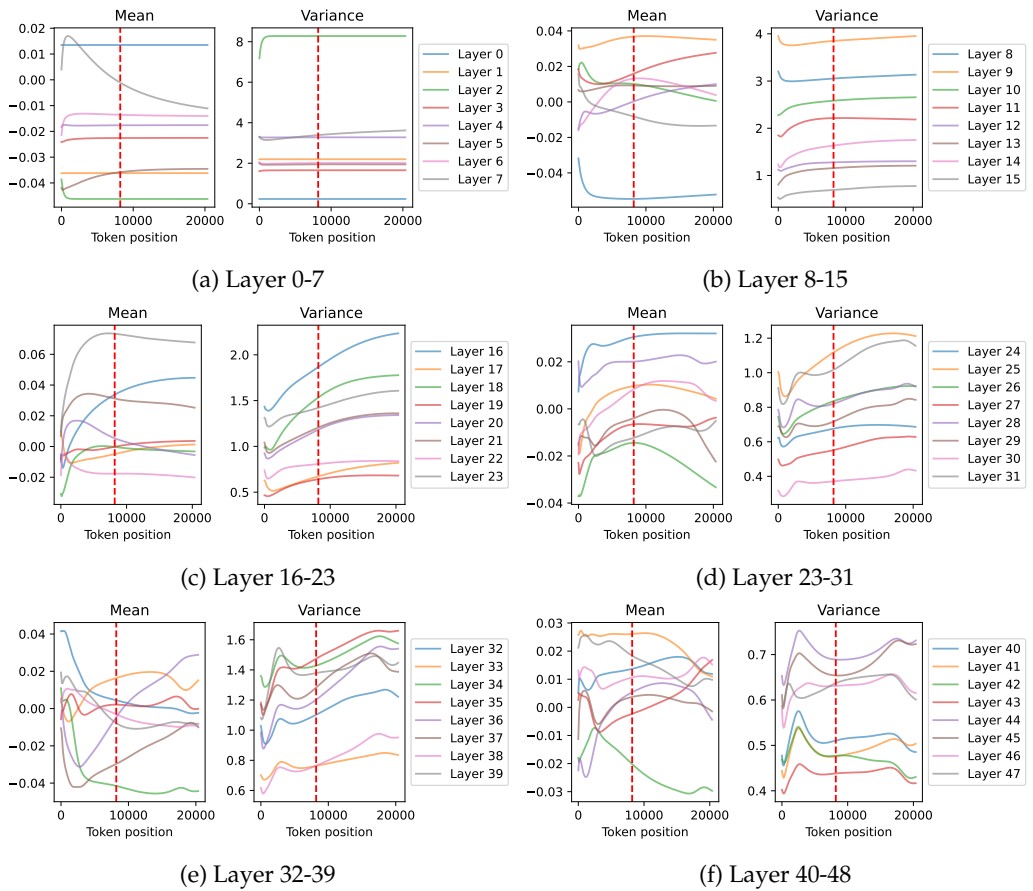

(a) Layer 0-7        (b) Layer 8-15

(c) Layer 16-23        (d) Layer 23-31

(e) Layer 32-39        (f) Layer 40-48

Figure 17: The mean and variance of the convolutional states (the representation of the last four tokens) of each layer in Mamba-2 370M, computed on the "newlines" prompt. We can see that the mean and variance are visibly more stable than the recurrent state.

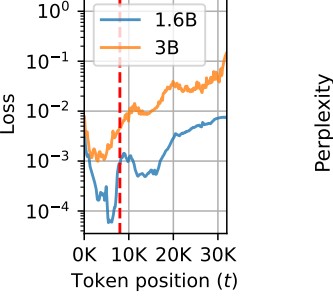

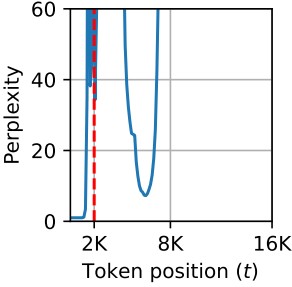

(a) Loss of RWKV-6 series on the "newlines" prompt as a function of time.

(b) The perplexity of HGRN-2 1.3B on the "newlines" prompt as a function of time.

