# OpenReview forum: "Stuffed Mamba: Oversized States Lead to the Inability to Forget"
_colmweb.org/COLM/2025/Conference — COLM 2025_

### Official Review · Reviewer_cVoT · 2025-05-03

**Rating:** 7
**Confidence:** 3
**Ethics Flag:** 1

**Summary:**

The paper analyses in depth the problem of Mamba where the performance drops drastically beyond context length. The paper made the following contributions in the understanding and improvement of this issue:
- In section 3, the paper exhibits various evidence that the model does not learn to forget and this behaviour explains the rapid decrease when the context exceeds the training context length
- In section 4, the paper examines the root cause of this memory retention and expose the relationship between training length and number of parameters characterizing the states.

**Questions To Authors:**

It seems that the paper provides possible explanations to Mamba/RWKV problems pointed out at https://arxiv.org/abs/2410.04727 where authors observe that at large length, the next token prediction performance falls below Mamba/RWKV that does not have access to the context.

**Reasons To Accept:**

The paper made important observations for RNN/Mamba type of architecture for future improvements. The observation at least guides for state matrix design and the training process (train for larger context length) to improve Mamba type performance.

The paper has significant experiments to support its claim. To argue that the performance drop is because of the inability to forget requires significant amount of experiments and I argue that the paper performs well. For example, 3.2.1 shows that some tokens are never forgot with certain heads and 3.2.2 shows that simply re-introducing forgetting improves the performance. In general, I find the paper's claim insightful and are backed up with solid experiments.

**Reasons To Reject:**

It seems that the paper can be presented more consistently in structure. Section 3 and Section 4 both are insights that are grounded with experiments. So it seems a bit strange to have a separate experiments section that only talk about threshold, maybe a better naming or organization can help readers.

Minorly, line 277, there seems some error as Figure 9 does not have a second figure.

---

> ### Author Response · Authors · 2025-05-31
>
> Thank you very much for your precious feedback.
>
> **Structure regarding the Experiments section**
>
> We agree that having a separate “Experiments” section is a bit strange. We will consider renaming Section 5 (“Experiments”) to “Forget and Recall Thresholds” to have a better parallel between this section and Sections 3 and 4.
>
> **Typo in line 277.**
>
> Thank you very much for this catch. “The second plot of” should be removed.
>
> **Possible Explanation for https://arxiv.org/pdf/2410.04727**
>
> Yes! Our findings can be viewed as a possible explanation for the observation in that paper. In fact, this work started exactly because we observed the same phenomenon as that paper.
>
> If there are any other aspects of our paper that you believe could be improved, please let us know.

---

### Official Review · Reviewer_DDLr · 2025-05-08

**Rating:** 8
**Confidence:** 5
**Ethics Flag:** 1

**Summary:**

This paper investigates the surprising failure of Mamba-based RNNs to extrapolate to longer sequences, despite their recurrent design. Contrary to the common belief that RNNs forget too much historical information, the authors argue that Mamba struggles because it forgets too little—retaining too much information due to oversized recurrent states. The paper provides compelling empirical evidence for this claim, showing that models trained on short contexts fail to learn effective forgetting mechanisms, leading to memory interference and degradation in performance beyond the training length. Notably, the authors introduce useful diagnostic concepts such as the Forget Threshold (minimum training length required to induce forgetting) and Maximum Recall Context Length (longest context length with reliable retrieval), offering valuable insights into the interplay between state size, training context, and memory retention.

I recommend a clear acceptance of this work.

**Reasons To Accept:**

- The paper proposes a very interesting hypothesis—that RNNs fail to generalize to long contexts not because they forget too much, but because they retain too much due to overparameterized states. This sharply contradicts the common belief that RNNs are prone to forget historical information

- The empirical evidence supporting this claim is thorough and convincing, including well-designed diagnostics (e.g., forget thresholds, recall lengths) and clear demonstrations of the failure modes and mitigation strategies.

**Reasons To Reject:**

While the empirical observations and analyses are insightful, the paper lacks a viable and practical solution to the length generalization problem. There appears to be an inherent trade-off between short-context performance and long-context generalization: pretraining on longer sequences tends to increase perplexity at short-context positions, and enforcing forgetting during training degrades short-context modeling. Moreover, pretraining beyond the identified Forget Threshold is computationally impractical at the 700M scale and beyond.

The proposed sliding window workaround preserves short-context performance but effectively discards all historical information. This undermines one of the key motivations for using RNNs in the first place—their ability to retain long-range dependencies in a compact manner—rendering the approach less compelling than simply using sliding-window attention.

Finally, the paper lacks an important baseline: post-training on longer sequences from an existing Mamba checkpoint for few tokens. This approach—similar to RoPE extension in Transformers—could potentially enable length generalization with minimal additional cost. Without evaluating such a baseline, it remains unclear whether the observed failure to generalize is a fundamental limitation of RNNs or merely a training artifact.

---

> ### Author Response · Authors · 2025-06-01
>
> Thank you very much for your precious review. We would like to make the following response.
>
> **Viable and Practical Solutions to the Tradeoff Between Short-Context and Long-Context Performance.**
>
> There is indeed an inherent trade-off between short-context performance and long-context generalization. This is a fundamental research question that is outside the scope of our work. We believe further improvements to the architecture and training approach can mitigate the issue. Interestingly, a similar (though not exactly the same) trade-off has also been observed with decoder-only Transformer models as well [1].
>
> **Pretraining beyond the identified Forget Threshold is computationally impractical.**
>
> To pretrain with the Forget Threshold identified in our paper is indeed a great challenge. Some techniques that we think are promising include recomputation, gradient truncation, sequence parallelism, gradient offloading, zero-order optimization, etc. Moreover, architectural improvements may also alleviate the need for very long contexts, such as designing update rules that are better at learning robust forgetting.
>
> **Sliding Window Mamba is Not Compelling.**
>
> The main purpose of the proposed sliding window method is to show that inducing forgetting can mitigate length generalization failure. Nonetheless, we would argue that this method has some merits compared to sliding window attention (SWA).
>
> - Firstly, it can be applied to any Mamba models (or other models whose recurrent state can be reformulated as a weighted sum) without additional training, even though the model was pretrained without a sliding window. In contrast, attention-based models that were not trained with SWA either require additional training or require modification to attention patterns and position embeddings to turn into a SWA model.
> - Secondly, the recurrent state may be considerably smaller than the KV cache in SWA, which means the model may have a lower memory cost.
>
> Extensively comparing these methods is outside the scope of our work.
>
> **The Lack of a Baseline: Context Length Extension by Post-Training**
>
> You are raising a very important point and we agree that such a baseline is very important. However, we found that directly training on longer sequences require large amounts of data for the model to adapt to the next context length. This can be partly explained by Figure 3. It shows that, after sufficient training, the majority of SSM heads in Mamba-2 have a memory retention strength ($\alpha_t$) whose cumulative products are very close to 0, resulting in vanishing gradient. This causes the model to struggle to learn to model long-distance dependencies. How to improve the data-efficiency in long-context extension for Mamba-2 remains an important future research direction.
>
> **References**
>
> [1] Buckman and Gelada. 2024. https://manifestai.com/articles/compute-optimal-context-size/

---

### Official Review · Reviewer_pm8h · 2025-05-13

**Rating:** 4
**Confidence:** 4
**Ethics Flag:** 1

**Summary:**

The paper offers empirical evidence seeking to confirm conditions under which Mamba-2 models "learn to forget."  The paper thus considers several retrieval-based experiments, aiming to demonstrate that there is a certain point during training that Mamba-2 models "learn to forget" and essentially overfit, and that if the training context length is sufficiently larger than Mamba-2's SSM dimension, the model retains the ability to forget.  Several experiments, model sizes, training context lengths, and SSM state sizes are presented to support the underlying hypotheses.

**Questions To Authors:**

- "We observe that forgetting occurs only when the training context length exceeds the state’s capacity to retain all information" <- Why would forgetting be correlated with context length?  Mamba's main innovation are the time-varying parameters, per time-step, so it effectively learns each time position during training.  Particularly since it is a first order process, learning should depend on the number of training steps, not on the training context length.

- "has a stable exponential memory decay (it converges to a constant value if the variables are fixed)" <- Citation or please prove this statement

- "Therefore, we expect RNNs of such form to have a good retrieval accuracy on the last k tokens, and tokens earlier than that are forgotten" <- Given Mamba's state-space equations, the argument that forgetting is dependent on length still doesn't make sense

"However, as the context length increases, Bs cannot be mutually orthogonal" <- Why?  Please explain.  If this is a mathematical argument, please try to prove the assertion.

"The retention strengths of earlier tokens are always smaller than those of more recent tokens." <- With your set up, you can prove this claim.  Please prove this claim.

"This implies that the model has not learned to forget information (by producing a smaller αj), but it still has decent language modeling capabilities because the information of 8K tokens is typically not enough to overload the memory" <- Why "8K tokens is typically not enough to overload the memory?" There is a break in logic between the experiment at the conclusions

In Equation 8, shouldn't r be w?

"which incurs minimal computational cost." <- This is not minimal, please change to neglible

"Similar observation can be found with any model size." <- Was this verified by the authors?

"This indicates that the model converges toward more retention and less forgetting" <- What learning schedule was used for pre-training, and how does it compare to the original Mamba-2 recipe?  If in the appendix, please forward reference.

Figure 8 is empirical evidence, more models seem warranted to support the underlying hypothesis.

"To determine whether the model has learned robust forgetting, we feed prompts with 1M tokens to the model and check if the model’s loss exceeds 2× the maximum loss within Ttrain tokens at any point." <- 2x seems very arbitrary

"The fact that the amount of information in the tokens exceeds the state’s capacity," <- the experiments are not convincing that this happening

" To save cost, we continue pre-training from three official checkpoints of Mamba-2 (130M, 370M, and 780M). They were pre-trained with 8K sequences. The other model configurations (36M, 47M, and 85M) are trained from scratch." <- Where are the training details?  As previously stated, if details are in the appendix, please forward reference.

"We can see that for each model size, there is a training length threshold, beyond which the model has much better length extrapolation, which supports our arguments discussed in Section 4.2." <- But why?  Is this an emergent ability?  I.e., if the x-axis on Figure (10) a-b were further extended, would the 16k and 64k trained models continue to have stable loss?

"The rightmost data point in the plot corresponds to Mamba-2 370M" <- What are the other data points?

By state size, the authors are referring to the internal Mamba-2 state dimension, correct?  E.g., d_state \in {64, 128} for official Mamba-2 checkpoints?

"This is because the amount of information in the context does not increase with its length" <- This contradicts many other statements in the paper

**Reasons To Accept:**

Mamba-2 models have garnered significant interest over the past two years.  Quantifying exactly how and when they retain (or lose) the ability to forget information has the potential to impact the performance of these models for both training and downstream application performance.

**Reasons To Reject:**

The underlying hypotheses and experiments need more refinement.  In particular, a discussion of the underlying additive state problem of RNNs like Mamba is warranted, as this is a known issue that has been addressed in other work (see [the Delta Rule section of the following](https://openreview.net/pdf?id=ayB1PACN5j) for more background).  However, exactly quantifying this for Mamba-2 models could still be a valuable contribution.  Yet, the experiments require further work; at times they lie at odds with one another.  In particular, the following two statements:
 > (1) Initially, the model demonstrates robust forgetting, retaining only the last k
> tokens and forgetting earlier ones.  However, as training progresses, the model’s ability to
>  forget diminishes while its recall of contextual information improves, resembling overfitting.
>  This suggests that the model increasingly attempts to retain all available information within
> the context. (2) We observe that forgetting occurs only when the training context length
> exceeds the state’s capacity to retain all information, forcing the model to forget less relevant
> details. Notably, larger states require longer training context lengths to effectively learn and
> implement forgetting.

Clearly, the two statements contradict one another; if (2) is true, (1) does not get to the point of overfitting.  Similarly, given (1), what if the state is especially small in the non-overfitting region (thus invalidating statement 2).  The paper has many such contradictory statements/hypotheses which require ironing out.  There are also statement which require verification and missing details.  Finally, while the underlying phenomena is interesting, what demonstratable does this have on real applications?  The last question should be addressed at length.

---

> ### Author Response · Authors · 2025-06-01
> **Rebuttal part 1/3**
>
> Thank you very much for the highly detailed review. We will try to address your remarks one by one. Bold text indicates "reason to reject"/question mentioned by you, and the text following each bold text is the corresponsing response.
>
> **Delta Rule.**
>
> The link you provided is not accessible for us. Can you please share an accessible link?
>
> Nonetheless, as you mentioned, improving Mamba-2 is valuable since it has been applied to several LLMs and has important practical values.
>
> **Contradicting statements.**
>
> We respectfully disagree with your claim that our paper contains contradicting statements.
> In response to the example you provided, we believe you have misunderstood the context of those two statements. Statement (1) describes the scenario when the training context lengths are too short for the state size (i.e., $T_\text{train} < T_\text{forget}$), which causes overfitting. Statement (2) describes the scenario when we increase the training context length beyond the Forget Threshold (i.e., $T_\text{train} \ge T_\text{forget}$), in which case overfitting does not occur. These two statements describe different scenarios, which do not occur at the same time. Thus, they are not contradicting statements.
>
> In response to your remark that “There are also statements which require verification and missing details”, can you kindly provide examples? We would be very glad to provide additional details/verification measures.
>
> **Real applications of the phenomena.**
>
> The findings are meant to provide insights for designing better Mamba-based models for long-context modeling. Thus, the application value of this paper lies in the takeaways, some important ones are:
> - To train a Mamba-2 with robust length generalization, one should use training lengths that grow linearly with the state size.
> - Length generalization failure can be caused by the lack of forgetting, instead of the lack of memory retention, as claimed in [1].
>
> **Mamba-2’s learning process should not depend on the training context length.**
>
> We respectfully disagree. The learning dynamics of Mamba-2 depend on the training context length because the output hidden representations at any layer and time step (i.e., $y_t$) depend on the input hidden representations in all previous time steps (i.e., $[u_1, \cdots, u_t]$). Thus, the training context length will affect the loss function, which influences the learned neural weights. In other words, the learned model weights (i.e., $A$, $W_x$, $W_\Delta$, $W_B$, etc. in Appendix A) are influenced by the training context length. Additionally, the time-varying variables (i.e., $\alpha_t$, $\overline B_t$ and $x_t$ in Eq. 2) are produced by these model weights, so they are also influenced by the training context length. Hence, the update rule and forgetting mechanism in Mamba-2 are influenced by the context length.
>
> **Mamba-2 has a stable exponential decay (converging to a constant value when variables are fixed).**
>
> The proof is as follows:
>
> Assuming fixed variables, we have $\alpha_i = \alpha, \overline B_i =  \overline B$ and $x_i = x$ for all $i=1,2,\cdots$, thus:
>
> $$
> h_t = \overline B x + \alpha \overline B x + \alpha ^ 2 \overline B x + \cdots + \alpha^{t-1}\overline Bx=\left(\sum_{i=0}^{t-1} \alpha^i \right)\overline B x \underset{t\rightarrow \infty}{=} \frac{1}{1-\alpha}\overline Bx
> $$
> QED.
>
> This proof will be added to the revised paper.
>
> **Mamba’s state-space equations’ influence on forgetting.**
>
> Can you kindly clarify why Mamba’s state-space equations would undermine the observation that context length affects the forgetting mechanism? As mentioned above, the learned weights are influenced by the context length; the variables in the update rule ($\alpha_t, \overline B, x_t$ in Eq. 2) are influenced by the learned weights; and the actual forgetting mechanism of a trained Mamba is influenced by the update rule. Hence, we conclude that the forgetting mechanism in a trained Mamba model is influenced by the context length.
>
> **Why cannot $\overline B_i$ be mutually orthogonal?**
>
> This is because $\overline B_i$ has $N=128$ dimensions. Thus, when there exists more than $N$ of them ($[\overline B_1, \overline B_2, \cdots, \overline B_T] \text{ where } T > N$), they cannot be mutually orthogonal. This is a basic result in linear algebra. This concurs with our statement in Line 148 because the context length can be greater than $N$.
>
> > Remark: We have mistakenly left out the `\overline` in Line 148. This will be fixed in the future.
>
> **References**
>
> [1] Ben-Kish. 2024. DeciMamba: Exploring the Length Extrapolation Potential of Mamba

---

> > ### Comment · Reviewer_pm8h · 2025-06-06
> > **Reply to Rebuttal part 1/3**
> >
> > > Mamba-2 has a stable exponential decay (converging to a constant value when variables are fixed).
> >
> > This is incorrect, "converging to a constant value when variables are fixed" does not imply "Mamba-2 has a stable exponential decay."  To be clear, when the values are fixed, i.e., Mamba is no longer time-varying, you get S5 models [1], the stability of which is derived in [2].  However, a relevant source for general stability (without downgrading to time-invariance) of Mamba SSMs would be [3].
> >
> >
> > ## References
> > [1] Jimmy TH Smith, Andrew Warrington, and Scott W Linderman. “Simplified State Space Layers for Sequence
> > Modeling”. In: The International Conference on Learning Representations (ICLR). 2023.
> >
> > [2] Orvieto, Antonio, et al. "Resurrecting recurrent neural networks for long sequences." International Conference on Machine Learning. PMLR, 2023.
> >
> > [3] Halloran, John Timothy, Manbir S. Gulati, and Paul F. Roysdon. "Mamba state-space models can be strong downstream learners." (2024).

---

> > > ### Author Response · Authors · 2025-06-07
> > >
> > > Thank you for clarifying the concern. We believe the wording involving "stable" and "fixed" is ambiguous. Hence, we will change this sentence to "Mamba-2's hidden state is bounded when the variables are bounded". Would that address your concern?
> > >
> > > Additionally, we respectfully request that you comment on whether your other concerns have been addressed. Such comments would be greatly valuable for us in improving the paper. Thank you very much in advance.

---

> > > > ### Comment · Reviewer_pm8h · 2025-06-09
> > > > **Reply**
> > > >
> > > > Thank you for the response.  The proposed change obfuscates the original statement.  Please see my original response, it's just a matter of citing the correct result when describing stability.
> > > >
> > > > > Mamba-2’s learning process should not depend on the training context length.
> > > >
> > > > > Mamba’s state-space equations’ influence on forgetting.
> > > >
> > > > There are two potentially confusing narratives being addressed: 1) that the Mamba architectures maintain a compressed state (as described in the paper), and (2) the inability to forget earlier tokens (which is a consequence of the loop unrolling equation of Section 3.1 in the Mamba-2 paper, that the authors leverage for their results).  (1) is a common narrative, and means Mamba SSMs are first order processes, which of course lead to linear inference complexity in the context-length.  I understand the authors are referring to (2) when stating "Hence, the update rule and forgetting mechanism in Mamba-2 are influenced by the context length," being clear about this in the second paragraph of the intro will help avoid potential confusion.
> > > >
> > > > Reading through the paper again, there still remained confusing points (even after having gone through the authors' responses), which warrants an editing pass.  I urge the authors to read through and adjust confusing/contradictory statements.  E.g.:
> > > > > We hypothesize that the inability to learn an effective forgetting mechanism is due to state
> > > > overparameterization—where the model’s state is excessively large, allowing it to minimize
> > > > language modeling loss without much forgetting.
> > > > Two key pieces of evidence support this hypothesis:
> > > > (1) Initially, the model demonstrates robust forgetting, retaining only the last k
> > > > tokens and forgetting earlier ones. However, as training progresses, the model’s ability to
> > > > forget diminishes while its recall of contextual information improves, resembling overfitting.
> > > > This suggests that the model increasingly attempts to retain all available information within
> > > > the context.
> > > > (2) We observe that forgetting occurs only when the training context length
> > > > exceeds the state’s capacity to retain all information, forcing the model to forget less relevant
> > > > details. Notably, larger states require longer training context lengths to effectively learn and
> > > > implement forgetting.
> > > >
> > > > These two sentences should be decoupled/disambiguated.  Read it: (1) states the model initially demonstrates robust forgetting, then (2) states a condition for which forgetting only occurs.  There is certainly a way to state what the authors mean without contradictions.
> > > >
> > > > # Greater placement within existing work
> > > > There is a significant portion of existing related work that requires more discussion to differentiate/situate the contributions of the work.  As pointed out by Reviewer rF9s25, one of these is LongMamba.
> > > >
> > > > This statement:
> > > > > We first emphasize that LongMamba [1] is a concurrent work because it was published after COLM’s submission deadline.
> > > >
> > > > is also contradicted by the fact that LongMamba is used in the paper.  At a bare minimum, a discussion of the differences between the presented work and LongMamba is necessary.
> > > >
> > > > The following is the corrected link from my review:
> > > > https://www.arxiv.org/pdf/2503.14456
> > > >
> > > > This is the RWKV-7 paper--I understand, this was very briefly cited in the submission--which I've included to point to their background section.  The ability to forget in first-order RNNs has been addressed in recent work, but these relations are not adequately covered in the paper.  I.e.:
> > > >
> > > > > Thus, some conclusions/insights may apply to other architectures. We leave such exhaustive ablation studies for future work.
> > > >
> > > > Addressing this is necessary to situate the contributions next to existing work (especially wrt Gated Delta Networks, which would have been an ideal model to conversely explore effective forgetting in models designed with that in mind).
> > > >
> > > > # Other questions
> > > > > The training details are reported in Appendix F.1. We will add a reference to this sentence to clarify this. The training recipe is quite similar to the original Mamba-2 recipe, with some slight variations that we believe do not affect our conclusions. Extensive ablation studies on the influence of the training recipe is interesting, but out of the scope of this work.
> > > >
> > > > Please add answers to the following in the paper: What are the number of training epochs?  What were the total number of training tokens?
> > > >
> > > > For Figure 1:
> > > > > We find that Mamba-2 (except for the smaller 130M checkpoint) has near-perfect retrieval accuracy within 8K tokens, but poor or even zero accuracy on sequences longer than 16K, regardless of model sizes.
> > > >
> > > > The models are trained on 8k data, what are the alpha_t values set to in these experiments for t > 8k?

---

> ### Author Response · Authors · 2025-06-01
> **Rebuttal part 2/3**
>
> **Why are the retention strengths of earlier tokens always smaller than those of more recent tokens?**
>
> The retention strength of the $i$-th token at time step $t$ is defined as $\alpha_{i:t}=\prod_{j=i+1}^t \alpha_j$, since $\alpha_i\in(0, 1)$, we have:
> $$
> \alpha_{i:t}=\alpha_i \cdot \alpha_{i+1:t} < \alpha_{i+1:t}
> $$
> for any $t$ and $i$. Thus, the retention strengths are monotonically increasing. QED.
>
> > Remark: In Line 113, we defined “memory retention strength” as $\alpha_t$, which might be misleading due to the similar name, thus, we will change this term in the future.
>
> We will add this proof in the revised paper.
>
> **Why are 8K tokens not enough to overload the memory?**
>
> Here, we are claiming that overloading the memory would cause the model to learn to produce smaller memory decay terms ($\alpha_t$), as argued in Section 3.1. Moreover, we are stating that the 8K context length is insufficient for the model sizes of the official Mamba-2 checkpoints (i.e, those we investigated in Section 3). In later experiments (Sections 5.1 and 5.2), we showed that the models of such sizes require greater than 8K to learn forgetting. Hence, our experiments support this claim.
>
> **Typos in Equation 8**
>
> Yes, it is a typo, the $w$ in Line 176 should be replaced with $r$.
>
> **The usage of “minimal” versus “negligible”.**
>
> Nice catch, we will change it to negligible.
>
> **"Similar observation can be found with any model size."**
>
> Yes, this is empirically verified, and we are reporting the results for one model size to save space.
>
> **"This indicates that the model converges toward more retention and less forgetting" <- What learning schedule was used for pre-training, and how does it compare to the original Mamba-2 recipe?**
>
> The training details are reported in Appendix F.1. We will add a reference to this sentence to clarify this. The training recipe is quite similar to the original Mamba-2 recipe, with some slight variations that we believe do not affect our conclusions. Extensive ablation studies on the influence of the training recipe is interesting, but out of the scope of this work.
>
> **Figure 8 is empirical evidence, more models seem warranted to support the underlying hypothesis.**
>
> We will add the result of Mamba-2 130M and 780M to the Appendix. This phenomenon is observed across different model sizes in our experiments as long as the training context length falls below the Forget Threshold (defined in Section 5).
>
> **"To determine whether the model has learned robust forgetting, we feed prompts with 1M tokens to the model and check if the model’s loss exceeds 2× the maximum loss within Ttrain tokens at any point." <- 2x seems very arbitrary**
>
> The choice of 2x was determined such that it can reliably capture severe length generalization failure while ensuring that fluctuations are not mistakenly regarded as failures. While we acknowledge that this value is an arbitrary choice, the conclusions are not very sensitive to the actual value (using 1.5x would give exactly the same conclusions). This is because at such huge context lengths, models that are unable to forget earlier tokens have very large loss values, so models with robust length generalization have a stable loss.
>
> **"The fact that the amount of information in the tokens exceeds the state’s capacity," <- the experiments are not convincing that this happening**
>
> This sentence describes the conclusion from the previous section (Section 4.2), which is empirically validated in Sections 5.1 and 5.2.
>
> **"To save cost, we continue pre-training from three official checkpoints of Mamba-2 (130M, 370M, and 780M). They were pre-trained with 8K sequences. The other model configurations (36M, 47M, and 85M) are trained from scratch." <- Where are the training details? As previously stated, if details are in the appendix, please forward reference.**
>
> As mentioned in Section 5 (Line 253), the training details are given in Appendix F.
>
> **"We can see that for each model size, there is a training length threshold, beyond which the model has much better length extrapolation, which supports our arguments discussed in Section 4.2." <- But why? Is this an emergent ability? I.e., if the x-axis on Figure (10) a-b were further extended, would the 16k and 64k trained models continue to have stable loss?**
>
> Yes, this is an emergent behavior when we scale up the training context length, and the reason for this emergence is discussed in Sections 4 and 4.2. We have empirically verified that the models in Figure 10 (a and b) have stable loss up to 10M tokens.
>
> **"The rightmost data point in the plot corresponds to Mamba-2 370M" <- What are the other data points?**
>
> The other data points correspond to the smaller model sizes listed in the “Models” paragraph in Section 5, which are 36M, 47M, 85M, and 130M.

---

> ### Author Response · Authors · 2025-06-01
> **Rebuttal part 3/3**
>
> **By state size, the authors are referring to the internal Mamba-2 state dimension, correct? E.g., d_state \in {64, 128} for official Mamba-2 checkpoints?**
>
> As defined in Section 4.2 and Line 272, the state size is $N_S$, which is the dimensionality of all states (i.e., $h_t$) at any time step. There are $H$ of them in each layer, where $H$ is the number of SSM heads. In this paper, we use M parameters as the measurement unit for $N_S$.
>
> **"This is because the amount of information in the context does not increase with its length" <- This contradicts many other statements in the paper**
>
> This does not contradict other statements in the paper because this statement is describing the prompts used in the passkey retrieval task, and not the other kinds of contexts. The concrete prompt in passkey retrieval is provided in Appendix B.1. In this prompt, a 5-digit passkey is located inside a large context of repeated texts. When increasing the context length, we just increase the number of repetitions, and there is no additional content. Hence, the amount of information is essentially independent of the context length. This prompt is adopted from the original authors of the passkey retrieval task [2].
>
> **References**
>
> [2] Mohtashami et al. 2023. Random-Access Infinite Context Length for Transformers.

---

> > ### Comment · Reviewer_pm8h · 2025-06-06
> > **Response to Rebuttal part 3/3**
> >
> > I see, I thank the authors for clarifying.  I recommend adding this point to the main text (maybe as a footnote).

---

> > > ### Author Response · Authors · 2025-06-08
> > >
> > > Thank you for the suggestion. We will add this point to the main text as you have recommended.
> > >
> > > We kindly ask whether there are any other concerns left unaddressed. If not, we respectfully hope that you can adjust your scores to reflect the resolution.
> > >
> > > Thank you very much for your time.

---

> ### Author Response · Authors · 2025-06-09
>
> Thank you very much for your response.
>
> # Influence of Context Length on Mamba-2's Update Rule
>
> When we say that Mamba-2's forgetting mechanism is influenced by the training context length, we mean that the data-dependent variables (specifically, the memory decay multiplier $\alpha_t$) are influenced by the training context length. This is true since the value of the neural weights that generate $\alpha_t$ are dependent on the training context length.
>
> As you have suggested, we will make this more explicit in the introduction to avoid confusion.
>
> ## Disambiguating Statements
>
> We agree that we can disambiguate the paragraph you mentioned further by explicitly stating that the two statements describe two different scenarios. Thus, we will revise this paragraph in the updated paper.
>
> # Greater Placement within Existing Works
>
> Regarding RWKV-7, which is a concurrent work, we would like to emphasize that our analysis is limited to recurrent architectures whose update rule can be written as an element-wise affine operator (Mamba-2, Mamba-1, RWKV-6, HGRN-2, etc.). RWKV-7 and DeltaNet are out of the scope of our paper.
>
> ## LongMamba
>
> Your claim regarding LongMamba is false. The LongMamba [1] paper referred to in the response to Reviewer rF9s25 is not the same as the one appearing in the paper [2] (which is a GitHub repository without any paper).
>
> # Other Questions 1
>
> Thank you for these questions. The number of training tokens is 50 times the model size, and we only train for 1 epoch (no training data repetition). We will add these details to the paper.
>
> # Other Questions 2
>
> Regarding $\alpha_t$ values for $t \gt 8K$, each $\alpha_t$ is a function of the $t$ input hidden state $u_t$ (please see Eq. 4). Mamba-2 does not have positional encodings, so this is well defined, and we do not make any modifications to the model.
>
> Once again, thank you very much for your precious time. If there are more concerns, we are very happy to address them through further discussions.
>
> # References
>
> [1] Ye et al. 2025. LongMamba: Enhancing Mamba's Long Context Capabilities via Training-Free Receptive Field Enlargement
>
> [2] https://github.com/jzhang38/LongMamba

---

### Official Review · Reviewer_rF9s · 2025-05-25

**Rating:** 5
**Confidence:** 4
**Ethics Flag:** 1

**Summary:**

This work investigates the forgetting behavior of Mamba-based models and shows that they struggle to effectively forget earlier tokens, even with built-in forgetting mechanisms. It demonstrates that the minimum training length required for the model to learn forgetting scales linearly with the state size, while the maximum context length for accurate retrieval scales exponentially with the state size. These observations may provide insights for the community to better understand Mamba's behavior.

**Questions To Authors:**

My questions have been included in the weakness section. I'm willing to adjust my scores if my concerns are properly addressed.

**Reasons To Accept:**

1. The paper is clearly written and easy to follow.

2. The finding regarding the relationship between minimum training length and state size is interesting and novel.

3. While not entirely new, the overall observations could enhance the community’s understanding of Mamba and other linear attention models.

**Reasons To Reject:**

1. The concept and causes of forgetting have been analyzed in previous works, e.g., DeciMamba and LongMamba. This work seems to provide a similar analysis of memory decay factors and then uses retrieval performance as a proxy to quantify forgetting. If the authors aim to emphasize the forgetting aspect, a deeper or lower-level analysis of the forgetting mechanism is expected, in addition to the empirical study similar to previous works.

2. When fitting the scaling laws in Figures 9 and 11, too few data points are used, making the conclusions less convincing.

3. The findings of this work are not entirely new. For example, the strong memory retention of early tokens, the relationship between state size and memorable context length, and the difficulty of extrapolating beyond the training length have been discussed in previous works like LongMamba and DeciMamba.

4. The authors mention that *"Mamba-2 with 370M parameters achieves near-perfect retrieval on a 256K context length, outperforming similarly sized transformer models,"* but I did not find this set of experiments. For this comparison, is ROPE extension applied to transformers to ensure the performance gap is not due to improper handling of other design factors?

5. More linear attention models beyond Mamba/Mamba2 are highly likely to be included in the discussion.

---

> ### Author Response · Authors · 2025-06-02
>
> Thank you very much for your time in reviewing.
>
> **Novelty of this paper and comparison to existing works.**
>
> We first emphasize that LongMamba [1] is a concurrent work because it was published after COLM’s submission deadline. Moreover, our work differs from existing works involving length generalization of Mamba-based models in the following aspects:
> - **Relationship between memory decay and length generalization**: In both DeciMamba and LongMamba, it is argued that too much memory decay (i.e., forgetting too much) is the cause of length generalization failure. In contrast, we show that the failure is caused by too little memory decay (i.e., the inability to forget).
> - **Length generalization ability**: Existing work in length generalization methods, such as DeciMamba [2] and LongMamba [1], can only extrapolate up to 40-50K tokens.
> - **Relationship between state size and memorable context length**: Existing works (e.g., [4]) that investigate the relationship between state size and memory are limited to toy tasks, and we are the first to establish this relationship with language modeling training.
>
> **Figures 9 and 11 have too few data points.**
>
> We agree that more data points would make the conclusions more convincing, thus, we have conducted additional experiments (with model sizes of 15M, 25M, 55M, and 100M). The new results (combined with previous results) are reported below, and they show that our established relationships/conclusions still hold. These results will be added to the revised paper.
>
> | Model size | State size ($N_S$) | Forget Threshold ($T_\text{forget}$) | Max. Recall Len. ($T_\text{recall}$) |
> | --- | --- | --- | --- |
> | 15M | 393K | 512 | 512 |
> | 25M | 786K | 768 | 768 |
> | 36M | 1.05M | 1K | 1K |
> | 47M | 1.57M | 2K | 2K |
> | 55M | 2.36M | 4K | 3K |
> | 85M | 2.95M | 8K | 4K |
> | 100M | 3.54M | 12K | 8K |
> | 130M | 4.72M | 16K | 16K |
> | 370M | 12.58M | 64K | 256K |
>
> **The result of near-perfect retrieval on a 256K context length.**
>
> This result is presented in Figure 9 (the rightmost point). “Max Recall Len.” (i.e., $T_\text{recall}$, defined in Section 4.3) refers to the context length at which the model has more than 95% accuracy in passkey retrieval, and the rightmost data point in Figure 9 corresponds to the 370M Mamba-2 model with near-perfect accuracy at 256K context lengths.
>
> Additionally, you mentioned “RoPE extension applied to transformer”. We are unsure about what the concern is. Would you kindly clarify?
>
> **More linear attention models beyond Mamba/Mamba2 are highly likely to be included in the discussion.**
>
> We are not sure what your concern is. We have presented results involving the length generalization failure of RWKV-6 and HGRN-2 in Appendix H. Our analysis about the importance of forgetting is not specific to Mamba, so it should be generalizable to other kinds of linear attention models. Empirical validation with all linear attention variants is impractical, so we have focused on Mamba (which is the most widely used linear attention variant at the moment).
>
> **References:**
>
> [1] Ye et al. 2025. LongMamba: Enhancing Mamba's Long Context Capabilities via Training-Free Receptive Field Enlargement
>
> [2] Ben-Kish et al. 2024. DeciMamba: Exploring the Length Extrapolation Potential of Mamba
>
> [3] Yang et al. 2024. Gated Linear Attention Transformers with Hardware-Efficient Training
>
> [4] Arora et al. 2024. Simple linear attention language models balance the recall-throughput tradeoff.

---

> > ### Comment · Reviewer_rF9s · 2025-06-09
> >
> > I thank the authors for providing the response. Some of my concerns are addressed, e.g., regarding more linear attention. However, I still have the following concerns:
> >
> > 1. If the authors wish to claim a different conclusion from DeciMamba/LongMamba (i.e., the issue of too little or too much memory decay mentioned by the authors), they should provide analysis and experiments to explain what leads to the differing conclusions and what connections exist between the works.
> >
> > In practice, both works mitigate excessive memory decay through token pruning. The authors should analyze the impact of token pruning on the "forgetting" effect in this work.
> >
> > 2. My question actually concerns the claim "outperforming similarly sized transformer models," as I did not see results for transformer models of comparable size. As a follow-up, if the authors did observe such results, did they adjust the base value of ROPE for these transformer models to ensure a fair comparison?
> >
> > I tend to keep my rating for now. Addressing the above concerns could strengthen this manuscript.

---

> ### Author Response · Authors · 2025-06-09
>
> Thank you very much for the response.
>
> Our response to the two concerns is as follows:
>
> 1. We are not making a conclusion that contradicts the conclusions from DeciMamba. Instead, we are providing another explanation for the length extrapolation failure. In fact, both too much and too little memory decay can lead to length generalization failure. However, in DeciMamba, they only considered the scenario where there is too much decay. Regarding token pruning, our analysis in Section 3.2.2 is already an analysis of token pruning on the "forgetting" effect. Sliding window is equivalent to pruning all tokens outside of the last $r$ tokens ($r$ is the window size). The conclusion is that token pruning induces forgetting, which mitigates the issue caused by the inability to forget.
> 2. Our claim that "Mamba-2 outperforms similarly sized transformer models" is based on publicly released transformer-based models. To the best of our knowledge, there are no publicly released transformer-based models of 370M parameters with such a strong performance on the passkey retrieval task. We are not claiming that transformer-based models cannot achieve this performance, but this is an impressive performance considering the great inference efficiency of Mamba-2.
>
>
> Thank you for your time, if there are any more concerns, please let us know.

---

### Decision · Program_Chairs · 2025-07-08

**Decision:**

Accept

**Comment:**

Maybe accept, if there is room.

This analysis paper makes and defends the interesting claim that Mamba suffers at long context length for a specific pair of reasons: because it not good at forgetting at those lengths, because of the limited context lengths at which it is trained.

There are split opinions by reviewers on this paper.   Supportive reviewers were enthusiastic and found the paper's proposal and development of a "Forget Threshold" and "Maximum Recall Context Length" (and the associated experimental data) to be valuable conceptual contributions for understanding the limitations of the Mamba architecture. AC agrees these seem like valuable contributions.

However, the main unresolved objection by negative reviewers is that the paper is insufficiently grounded in current work, in particular that both RWKV-7 and DeltaNet have advocated a "delta rule" that improves long context behavior at the cost of additional computational complexity, and that the simultaneous discoveries by LongMamba are not discussed.  The authors do not share these concerns because they deem those methods' (more computationally complex) non-elementwise update rule as "out of scope" for their paper, and also point out that LongMamba was not published at time of submission.

Both reviewer's questions and the author's point of view are reasonable.  I support accepting the paper if there is room.

If there is room to accept this paper, I urge the authors to add a more in-depth discussion of the delta rule, even if the main point is to help readers understand how it differs, and why it is out of scope for the authors' analysis.  And despite the timing of the simultaneous submission, I would ask the authors to consider adding a discussion of LongMamba and whether the authors' findings are consistent or not with the findings of that other concurrent work.  If the needed discussions are lengthy, it would be appropriate for supplemental material.